# Multifunctional Plasmon-Induced Transparency Devices Based on Hybrid Metamaterial-Waveguide Systems

**DOI:** 10.3390/nano12193273

**Published:** 2022-09-21

**Authors:** Hongting Chen, Zhaojian Zhang, Xiao Zhang, Yunxin Han, Zigang Zhou, Junbo Yang

**Affiliations:** 1College of Sciences, Southwest University of Science and Technology, Mianyang 621010, China; 2College of Sciences, National University of Defense Technology, Changsha 410073, China; 3College of Advanced Interdisciplinary Studies, National University of Defense Technology, Changsha 410073, China

**Keywords:** plasmon-induced transparency, metamaterial, graphene, Friedrich–Wintge bound state in continuum, guided mode

## Abstract

In this paper, we design a multifunctional micro-nano device with a hybrid metamaterial-waveguide system, which leads to a triple plasmon-induced transparency (PIT). The formation mechanisms of the three transparent peaks have their own unique characteristics. First, PIT-I can be switched into the BIC (Friedrich–Wintge bound state in continuum), and the quality factors (Q-factors) of the transparency window of PIT-I are increased during the process. Second, PIT-II comes from near-field coupling between two bright modes. Third, PIT-III is generated by the near-field coupling between a low-Q broadband bright mode and a high-Q narrowband guide mode, which also has a high-Q transparent window due to the guide mode. The triple-PIT described above can be dynamically tuned by the gate voltage of the graphene, particularly for the dynamic tuning of the Q values of PIT-I and PIT-III. Based on the high Q value of the transparent window, our proposed structure can be used for highly sensitive refractive index sensors or devices with prominent slow light effects.

## 1. Introduction

In plasmonic metamaterials, periodic nanostructures such as split-ring resonators (SRRs) and cut wires (CWs) can support resonances when the incident light is coupled to the surface of the nanostructure. The direct interaction between incident waves and surface electrons of the nanostructure causes the collective oscillation of electrons on the surface of the materials [1,2], which is what we call the superradiant mode (or bright mode). If the mode cannot be directly coupled with incident waves, it is called the subradiant mode (or dark mode). When bright and dark modes exist simultaneously, the coupling with destructive interference between the bright and dark modes produces the phenomenon of optical transparency, and this is the physical mechanism of plasmon-induced transparency (PIT) [3]. In addition to the physical mechanisms mentioned above, there is another physical mechanism for generating the PIT effect through the coupling between two bright modes [4]. It is known that the PIT effect based on metamaterials was initially proposed by Zhang et al. in 2008 [5], which can be regarded as an ideal alternative to the electromagnetically induced transparency (EIT) effect since the experimental conditions for EIT have been difficult [6]. The rational structural design of metamaterials can achieve the PIT effect that can possess the same functions as the EIT effect, i.e., a slow light effect in the transparent window. In addition, thanks to the introduction of tunable materials such as semiconductors [7], phase change materials [8], and graphene [9,10], various structures have been proposed to realize tunable PIT functional devices, such as optical switches [11,12], sensing [13,14,15], filtering [16,17], etc.

We mentioned that one important feature of the PIT effect is the slow light effect near the transparency window, and the strength of the slow light effect is related to the quality factors (Q-factors) of the transparent window [18]. Therefore, designing a PIT with a high-Q is the key to achieving a strong slow light effect; meanwhile, this can also benefit high-sensitive refractive index sensing and narrow-band filtering. On the one hand, the Q-factors are directly related to the loss of the material [19]. Therefore, we can consider using low-loss material for the nanostructure. On the other hand, we can obtain high-Q resonances by some resonant effects. The bound state in the continuum (BIC) is one approach to achieving high-Q resonances [20]. Theoretically, the ideal BIC would have infinite Q due to the energy in the resonator being strongly localized and not leaking into the continuum [21]. Therefore, a high-Q PIT can be constructed in the above two ways. Several previous studies have discussed the relation between BIC and PIT [22,23,24]. The conclusion reveals that we can obtain quasi-BIC by breaking the symmetry of the symmetry-protected BIC or tuning some parameters of Friedrich–Wintgen (FW) BIC to make it slightly deviate from the critical coupling condition. The FW quasi-BIC we discussed above is the high-Q Fano resonance generated by the coupling of two resonators. Specifically, PIT can be understood as a particular case of Fano resonance (q=0 or ±∞) when the two resonators are spectrally matched [25]. Based on such a principle, we can connect the PIT effect with the FW quasi-BIC effect and realize the free switching between them by tuning resonant frequencies of coupled resonances.

In this paper, we propose a triple-PIT micro-nano functional device based on a hybrid metamaterial-waveguide system, which employs the two methods mentioned earlier for constructing high-Q PIT. The PIT-I generated by the near-field coupling between outer split-ring resonators (OSRR) and rectangular nanoribbons (RECTs) can be switched into the BIC. PIT-II comes from near-field coupling between OSRR and inner split-ring resonators (ISRR). The PIT-III is generated by the near-field coupling between ISRR, RECTs, and a guide mode. The triple-PIT can be dynamically tuned by the gate voltage of the graphene, particularly for the dynamic tuning of the Q values of the transparent window. Therefore, our proposed structure can be used for highly sensitive refractive index sensors, slow light devices, multi-frequency optical switches, and narrowband filtering.

## 2. Structural Design

Our proposed overall structure is shown in Figure 1a, where Layer I is a refractive index matching layer with refractive index $n_{1}$ = 1.45 and height $h_{1}$ = 20 μm. Layer II is a waveguide layer with refractive index $n_{2}$ = 1.90 and height $h_{2}$ = 20 μm. Layer III is the substrate, and we assume the substrate thickness as semi-infinite in this simulation. Figure 1b shows the unit cell of the structure. We divided the whole structure into three parts: the first part was the OSRR, the second part was the ISRR, and the last part was the RECTs. The geometric parameters of the above structure were: *g* = 10 μm, $R_{1}$ = 53 μm, $R_{2}$ = 43 μm, $r_{1}$ = 35 μm, $r_{2}$ = 25 μm, *L* = 10 μm, *d* = 15 μm, and $\theta =40^{\circ }$. In this paper, the finite difference time domain method (FDTD) was used for numerical simulation. The periodic boundary conditions were used along the x-directions and y-directions, while the z-direction had perfectly matched absorbing layers. The structural units were connected through the metal wires, and changing the gate voltage supplied on the electrode could tune the Fermi energy level of graphene $E_{f}$.

In this structure, the temperature was set to T=300 K, the materials of OSRR and ISRR were metal aluminium (Al), and the thickness of OSRR and ISRR was set as *t* = 200 nm. The Drude model can characterize the optical properties of Al as follows: [26]
(1)$$\varepsilon_{Al}=\varepsilon_{\infty }-\frac{\omega_{p}^{2}}{\omega^{2}+i\omega \gamma }$$
where the plasma frequency $\omega_{p}=2.24\times 10^{16}$ rad/s, and the damping constant $\gamma =1.22\times 10^{14}$ rad/s.

As for the RECTs, we set their material as graphene with tunable conductivity to achieve dynamic tuning of the whole transmission spectrum. We modeled graphene with the 2D surface conductivity method in this work. For the surface conductivity of graphene, we can characterize it by the following equation [27]:(2)$$\sigma \left(\omega ,\Gamma ,E_{f},T\right)=\sigma_{\mathrm{intra}\;}\left(\omega ,\Gamma ,E_{f},T\right)+\sigma_{\mathrm{irrer}\;}\left(\omega ,\Gamma ,E_{f},T\right)$$
(3)$$\sigma_{\mathrm{intra}\;}\left(\omega ,\Gamma ,E_{f},T\right)=\frac{-ie^{2}}{\pi \hbar^{2}\left(\omega +i2\Gamma \right)}\int_{0}^{\infty }\xi \left(\frac{\partial f_{d}\left(\xi \right)}{\partial \xi }-\frac{\partial f_{d}\left(-\xi \right)}{\partial \xi }\right)d\xi$$
(4)$$\sigma_{\mathrm{inter}\;}\left(\omega ,\Gamma ,E_{f},T\right)=\frac{ie^{2}\left(\omega +i2\Gamma \right)}{\pi \hbar^{2}}\int_{0}^{\infty }\xi \left(\frac{\partial f_{d}\left(-\xi \right)-\partial f_{d}\left(\xi \right)}{{\left(\omega +i2\Gamma \right)}^{2}-4{\left(\xi /h\right)}^{2}}\right)d\xi$$
(5)$$f_{d}\left(\xi \right)=\frac{1}{\exp\left(\left(\xi -E_{f}\right)/k_{B}T\right)+1}$$
where Equation (Equation 2) shows that the total conductivity of graphene consists of two components, where $\sigma_{\mathrm{intra}\;}$ and $\sigma_{\mathrm{inter}}$ are the intraband and interband terms, respectively, ω represents the angular frequency, Γ represents the scattering rate, and the relation between the scattering rate and relaxation time is Γ=1/2τ. Furthermore, we can calculate the relaxation time by $\tau =\mu E_{f}/eV_{F}$ [28], where the carrier mobility can be $\mu =4\;m^{2}/\mathrm{Vs}$ [29]. $E_{f}$ represents the chemical potential energy of graphene, *e* is the charge of a single electron, and $V_{F}$ is the Fermi velocity, which we set to $V_{F}=10^{6}$ m/s. $k_{B}$ is the Boltzmann constant, and *ℏ* is the reduced Planck constant. Although the surface conductivity method mentioned above is only valid for monolayer graphene, we can still model multi-layer graphene by scaling the total conductivity by the number of layers. The conductivity of *N*-layer graphene is Nσ [30].

## 3. Results and Discussion

In order to analyze the physical mechanism of the triple-PIT, we first disregarded the waveguide layer and only considered the structure that includes the patterned structure and a dielectric substrate. As we mentioned earlier, the patterned structure is divided into three parts, which are the OSRR, ISRR, and RECTs. The transmission spectrum of each structure in Figure 2a–c shows that the OSRR, ISRR, and RECTs (Al with graphene) can be coupled with y-polarized normal incidence waves. Meanwhile, the corresponding distribution of |*E*| of the structure is shown in Figure 2e. The transmission curves and electric field distribution show that there are three resonances that are bright modes with y-polarized normal incidence. The ISRR or RECTs (Al with graphene) cannot be exciting as bright modes with x-polarized normal incidence. Therefore, our design structure is polarization-sensitive. Both the polarization and the method of incidence have a significant effect on the transmission spectrum of the structure. Therefore, all the simulations are simulated with y-polarized normal incidence. For the convenience of distinguishing the three modes, they are regarded as bright mode-I on OSRR, bright mode-II on RECTs, and bright mode-III on ISRR, respectively. We notice that the RECT (only graphene) in Figure 2c (i) cannot be coupled with incident waves because RECT (only graphene) are two discrete structures, and its geometric length in the polarization direction is not long enough to support resonances. The transmission curve (blue dashed line) also proves that it cannot be exciting as a bright mode. The two discrete graphene nanoribbons are connected through the part of the OSRR as shown in Figure 2c (ii). The surface current distribution of RECTs (Al with graphene) (ii) shown in Figure 2d indicated that the bright mode could be excited because the combined structure can resonate together via surface electric currents. The rest of our proposed RECTs are the cases shown in Figure 2c (ii) unless otherwise stated.

### 3.1. Single-PIT (quasi-BIC)

To investigate the physical mechanism of PIT-I, we only considered OSRR and RECTs, and here, their materials are Al. When the length of RECTs *d* is changed from 38 μm to 42 μm, we can tell from the distribution of $E_{z}$ in Figure 3c that the resonant structure at frequency $f_{1}$ is OSRR, the resonance at $f_{3}$ comes from RECTs, and both OSRR and RECTs resonate at $f_{2}$. Therefore, PIT-I is formed by coupling bright mode-I and bright mode-II. The peak’s linewidth gradually decreases when we further increase the *d*, and the generation condition of Friedrich–Wintgen BIC is met at *d* = 41 μm, corresponding to one mode with zero radiation rate, which means that the mode enters into BIC status and obtains a infinite Q factor (defined as Q = ω/2γ). With the further increase of *d*, the frequency of bright mode-II is red-shifted, resulting in the BIC switch to the quasi-BIC. Coupled mode theory (CMT) is empolyed to gain insight into the resonance mechanism for the PIT (quasi-BIC) switch to BIC. For the two-port system, the CMT can be written as [22,31]:
(6)$$\frac{d}{dt}\left[\begin{array}{c} a\\ b \end{array}\right]=i\left[\begin{array}{cc} \omega_{1}+i\gamma_{1} & k+i\gamma_{12}\\ k+i\gamma_{21} & \omega_{2}+i\gamma_{2} \end{array}\right]\cdot \left[\begin{array}{c} a\\ b \end{array}\right]+\left[\begin{array}{cc} g_{11} & g_{12}\\ g_{21} & g_{22} \end{array}\right]\left[\begin{array}{c} A_{1+}\\ B_{2+} \end{array}\right]$$
where *a* and *b* represent the amplitude of two modes, and $\omega_{1}$ and $\omega_{2}$ represent the angular frequency of the two resonators. $\gamma_{n}$ = $\gamma_{in}$ + $\gamma_{rn}$ is the decay rate of the resonators, and $\gamma_{n}$ includes the intrinsic loss $\gamma_{in}$ and radiative loss $\gamma_{rn}$. *k* is the direct coupling rate between two modes, while $\gamma_{12}$ and $\gamma_{21}$ are the coupling coefficient caused by the damping. In the resonance system, $g_{ij}$ represents the coupling between mode *i* and port *j* ((i,j∈[1,2]). The energy transfer relationship between the incident wave and the transparent wave can be written as follows:(7)$$\left[\begin{array}{c} A_{1-}\\ B_{2-} \end{array}\right]=\left[\begin{array}{cc} r & t\\ t & r \end{array}\right]\cdot \left[\begin{array}{c} A_{1+}\\ B_{2+} \end{array}\right]+\left[\begin{array}{cc} k_{11} & k_{12}\\ k_{21} & k_{22} \end{array}\right]\left[\begin{array}{c} a\\ b \end{array}\right]$$

$A_{1-}$ and $B_{2-}$ are the outgoing wave amplitude at port 1 and port 2, respectively. *r* and *t* represents the direct reflection and transmission coefficient when the nanostructure are not present in the transmission path between port1 and port 2. While $k_{ij}$ are the coupling between mode *i* and port *j*(i,j∈[1,2]). Considering the symmetry and energy conservation of the system, $\gamma_{12}$ = $\gamma_{21}$ = $\sqrt{\gamma_{1}\gamma_{2}}$, $g_{11}=g_{21}=k_{11}=k_{21}=\sqrt{\gamma_{r1}}$, $g_{22}=g_{12}=k_{22}=k_{12}=\sqrt{\gamma_{r2}}$. In the calculation, we ignore the intrinsic loss $\gamma_{in}$ to explain Figure 3a better. When the incident wave is input from port 1 to port 2, Equation (Equation 6) is solved as follows:(8)$$b=\frac{\left(ik-\gamma_{12}\right)\cdot \sqrt{\gamma_{1}}+\left(i\omega -\omega_{1}+\gamma_{1}\right)\cdot \sqrt{\gamma_{2}}}{\left(i\omega -i\omega_{1}+\gamma_{1}\right)\left(i\omega -i\omega_{2}+\gamma_{2}\right)-{\left(ik-\gamma_{12}\right)}^{2}}\cdot A_{1+}$$
(9)$$a=\frac{\left(ik-\gamma_{12}\right)}{\left(i\omega -i\omega_{1}+\gamma_{1}\right)}\cdot b+\frac{\sqrt{\gamma_{1}}}{\left(i\omega -i\omega_{1}+\gamma_{1}\right)}\cdot A_{1+}$$
(10)$$B_{2-}=tA_{1+}-\sqrt{\gamma_{1}}\cdot a-\sqrt{\gamma_{2}}\cdot b$$

Due to the direct transmission coefficient *T* = $B_{2-}/A_{1+}$ and *t* = 1, the theory results can be calculated by Equation (Equation 10) for the PIT (quasi-BIC) switch to BIC as shown in Figure 3b. For the FW-BIC, we can switch quasi-BIC to BIC by keeping the parameters of one oscillator unchanged while adjusting the frequency of another oscillator so that the generation condition of FW-BIC is met. The theoretical results from CMT can sufficiently describe the switch process between PIT and BIC, which confirm the physical mechanism behind the numerical results. However, we need to notice that material loss $\gamma_{in}$ is not considered in theoretical calculations. Thus, the transmission of the peak is smaller than the theoretical calculation.

### 3.2. Double-PIT

After discussing the physic mechanism of PIT-I, we introduced the ISRR to the structure. The OSRR and ISRR both interact with the RECTs because their resonant frequencies have spectral overlaps. Therefore, we set the material of RECTs as graphene, and the overall transmission curves can be dynamically tuned by the gate voltage of graphene. As shown in Figure 4a, a PIT is generated with only OSRR and ISRR. Figure 4c indicates that the resonance of dip 1 comes from OSRR, the resonance of dip 3 comes from ISRR, and both OSRR and ISRR resonate at peak 2. Therefore, the near-field coupling between bright mode-I and bright mode-III leads to PIT-II. Furthermore, the coupling between the three bright modes leads to the double-PIT shown in Figure 4b. The reason for the occurrence of double-PIT is that the frequency of bright mode-II is in the middle of bright mode-I and bright mode-III. Thus, they can be coupled with each other. The coupling also results in the frequency of peak 2 (whole structure) being blue-shifted compared with the situation of OSRR with ISRR. More importantly, Figure 4d shows that the PIT-I in Figure 4b remains consistent with Figure 3a, where the coupling of bright mode-I and bright mode-II leads to PIT-I.

We also investigated the tunability of the double-PIT by changing the Fermi energy level from 1.0 eV to 0.2 eV. We can learn from Equation (Equation 11) that response wavelength of graphene is inversely proportional to the $E_{f}$. Therefore, the decrease of the $E_{f}$ would cause the red-shift of the dip 2
(11)$$\lambda \propto \sqrt{\frac{2\pi^{2}\hbar cL}{\alpha_{0}E_{f}}}$$(blue arrow), as shown in Figure 5a. The reason why the frequency of dip 1 (red arrow) remains unchanged is that the resonance structure of dip 1 is metal Al, and it is not effected by the Fermi energy level. Moreover, we also noticed that the frequency of dip 3 had been slightly red-shifted when the $E_{f}$ was decreasing. It can be concluded that the bright mode-III represented by dip 3 is actually come from ISRR and RECTs, except that ISRR occupies the primary role in resonance. Thus, the red shift is slight, as shown in Figure 5b. The decrease in $E_{f}$ will cause the coupling effect between bright mode-II, bright mode-I, and bright mode-III to decrease. On the contrary, the coupling between bright mode-I and bright mode-III is increasing. It follows that the transmission curve of PIT-II is almost the same in the OSRR+ISRR, as shown in Figure 4a, when the $E_{f}$ is equal to 0.2 eV. To summarize, the change in $E_{f}$ would tune the transmission curve except for dip 1 in the situation of double-PIT. In particular, Figure 5a shows that peak 1 disappears when $E_{f}$ is equal to 0.4 eV. This suggests that the condition to produce FW-BICs is met when the frequencies of bright mode-I and bright-II are close to equal. Accordingly, the linewidth of peak 1 is decreasing to invisible with the closing to the BIC point. With the further decreasing of $E_{f}$, the frequency of bright mode-II is red-shifted, resulting in the BIC switch to the quasi-BIC, and this also indicates that BIC is indeed generated when $E_{f}$ is equal to 0.4 eV. In fact, the quasi-BIC is also a Fano resonance generated by coupling two low-Q modes. In addition, the PIT can be understood as a particular case of Fano interference, in which the *q* of Fano resonance is equal to 0 or ±∞. On this basis, the PIT-I is connected with BIC, and the free switching between PIT-I and BIC can be achieved, as shown in Figure 5c.

### 3.3. Triple-PIT (Guided Mode)

After achieving the Q-factor of PIT-I that can increase with the process that the PIT switches to the BIC, we consider adding the waveguide layer to construct triple-PIT. The Q-factor of guided mode is high because the material of the waveguide layer here is chosen as a dielectric, and meanwhile, the light is confined well in the waveguide with no lateral leakage. Combining the high-Q guided mode with the low-Q wideband bright mode can generate high-Q PIT. This method has also been reported in previous studies [32,33]. Typically, the guided mode cannot be excited because of the momentum mismatch between guided mode and electromagnetic waves from free space. However, the guided mode can be excited by the externally incident wave after putting a grating structure on the waveguided layer, providing the necessary momentum to couple the diffracted waves into the guided mode. Here, our metamaterial layer (patterned structure and refractive index match layer) replaces the grating role to provide momentum compensation to couple the incident waves into the guided mode.

As can be seen from Figure 6a, peak 3 and dip 4 were introduced to the transmission curves after adding the waveguided layer. It can be seen from the blue curve and distribution of |E| in Figure 6a that dip 4 represents the TE mode. Consequently, the near-field coupling between bright mode-III (ISRR + RECTs) and TE mode cause Fano resonance. However, we can change the geometric parameters of ISRR to make the bright mode-III and guided mode spectrally matched, and the Fano resonance can be switch to PIT-III, as shown in the red transmission curves. To further investigate whether the metamaterial layer can provide the momentum compensation to the waveguided layer. The excited of the guided mode needs to satisfy the energy dispersion for self-consistency [34,35,36]:
(12)$$\tan\left(kt_{w}\right)=\frac{k\left(\gamma_{c}+\gamma_{s}\right)}{k^{2}-\gamma_{c}\gamma_{s}},\left(TE-\;\mathrm{mode}\;\right)$$
(13)$$\tan\left(kt_{w}\right)=\frac{n_{w}^{2}k\left(n_{s}^{2}\gamma_{c}+n_{c}^{2}\gamma_{s}\right)}{n_{c}^{2}n_{s}^{2}k^{2}-n_{w}^{4}\gamma_{c}\gamma_{s}},\left(TM-\;\mathrm{mode}\;\right)$$
where $k=\sqrt{n_{w}^{2}k_{0}^{2}-\beta^{2}}$, $\gamma_{c}=\sqrt{\beta^{2}-n_{c}^{2}k_{0}^{2}}$, and $\gamma_{s}=\sqrt{\beta^{2}-n_{s}^{2}k_{0}^{2}}$. $t_{w}$ represents the thickness of the waveguide layer, which we take as 20 μm. $k_{0}$ is the free space wave vector, and it can be calculated by $k_{0}=2\pi /\lambda$. In our design, the necessary momentum compensation is provided by the metamaterial layer, which has a period $P_{x}$ = 110 μm, $P_{y}$ = 106 μm, so the phase-matching condition can be written as [37,38]:(14)$$\beta =k_{x-y}+\beta_{x}i+\beta_{y}j$$

The $k_{x-y}$ represents the wave vector component of the incident plane wave in the xoy plane. In our design, the source is vertically incident from free space, and there is no wave vector component in the xoy plane. Thus, for a plane wave excitation with polarization direction *x* or *y*, the propagation constant is:(15)$$\beta =\frac{2\pi }{P_{x}}x\;\mathrm{or}\;\beta =\frac{2\pi }{P_{y}}y$$

By inserting this propagation constant into Equation (Equation 12), the theoretical resonant wavelengths of the guided mode of our structure can be obtained. The results are shown in Figure 6b, which demonstrates that the theoretical resonant frequency is 1.841 THz, which is close to our simulation results *f* = 1.839 THz. Therefore, the data illustrate the consistency of our theoretical calculation results and simulations.

In order to further investigate the tunability of the triple-PIT, we changed the $E_{f}$ from 1.0 eV to 0.2 eV. Figure 7a,c shows that PIT-I can still switch to BIC with a decrease in $E_{f}$ when the guided mode was introduced. The slight red shift for PIT-II also exists in the same manner as the situation of double-PIT. As for the PIT-III, the transmission of peak 3 increases with the decrease in $E_{f}$, accompanied by the increase in FWHM. These results show that we can dynamically tune the triple-PIT. To demonstrate more directly the dynamic tuning role of graphene in this hybrid system, we show the reflection and absorption curves in Figure 7d,e, respectively. It is obvious that the absorption exists in the spectrum except for the frequency of the OSRR resonant, and the absorption mainly comes from graphene. Therefore, this phenomenon proves the interaction between RECT, ISRR, and the guided mode. These results are consistent with our early discussion. In addition, the Q-factor of PIT-I (quasi-BIC) is gradually increased with the decrease in $E_{f}$. On the contrary, the Q-factor of PIT-III decreases, which shows an opposite trend. The Q-factor of PIT-I increases because the frequency of bright mode-II is close to the frequency that can cause the occurrence of BIC. As shown in Figure 7b, the surface conductivity of graphene decreases with the decrease in $E_{f}$, and graphene tends to be similar to a semiconductor material at low $E_{f}$. Conversely, the graphene can be regarded as a metal material at high $E_{f}$. The deeper reason for this phenomenon is that the interaction between electromagnetic waves and material was strongly related to the free electron concentration. Meanwhile, the free electron concentration will change with the tuning of $E_{f}$. Thus, the coupling between bright mode-III and bright mode-II becomes weak, leading to the coupling between bright mode-III and guided mode becoming stronger when the $E_{f}$ decreases.

## 4. Application

Finally, we also demonstrated the potential application of our proposed structure after discussing the mechanisms. Considering the Q-factor of the transmission window of the triple-PIT, PIT-I and PIT-III have great value in sensing and slow light. To further improve the Q-factor of transmission window, we added the graphene layer to enhance the resonance and increase the carrier mobility μ to decrease the material loss. The layer of graphene is set as *N* = 7 in the following discussions.

### 4.1. Index Sensor

One of the most prominent applications for high-Q transparency is refractive index sensing. Here, the PIT-III is not included in the discussion because the guided mode is inside the structure, which means it is insensitive to changes in the environment. Figure 8a,b shows that the transmission and FWHM of PIT-I both decrease with the change in $E_{f}$ from 1.1 eV to 0.6 eV. Correspondingly, the Q-factor of the PIT-I transmission window gradually increases. Therefore, considering the large Q value and transmittance, only the case of $E_{f}$ = 0.8 eV is discussed. The results obtained from Figure 8c show that the PIT-I red-shifted with an increase in the refractive index of the outer environment when the $E_{f}$ was set as 0.8 eV. We can calculate the sensitivity of refractive index sensing by the formula S=Δf/Δn, which results in *S* = 0.30 THz/RIU. There is another parameter to evaluate the refractive index sensing performance, which is the FOM factor:(16)$$\mathrm{FOM}=\frac{S}{\mathrm{FWHM}}$$The FOM factor mainly characterizes the minimum refractive index change that can be detected by the refractive index sensing. The FOM increases from 14 to 16.6 due to the FWHM decreasing when the refractive index of the outer environment increases. The comparison in Table 1 indicated that the good performance in sensing. However, the performance of *S* loses to Refs. [39,40], especially Ref. [39]. The sensor in both references is designed with metal, which means the FOM can only be tuned by changing the structure. Our design sensor’s FOM can be improved by decreasing the $E_{f}$ while maintaining the parameters unchanged. In a word, we design a FOM tunable refractive index sensor.

### 4.2. Slow Light Device

One of the most prominent features of PIT is the slow light effect near the transmission window, which can effectively reduce the group velocity of light. We can measure the strength of the slow light effect by calculating the group delay, and the formula is as follows:(17)$$\tau =\frac{d\varphi }{d\omega }$$
where φ represents the phase shift, and ω represents the angular frequency. As shown in Figure 9a,c, the Q-factor of PIT-I changes from 52.5 to 38.4, and the corresponding group delay decreases from 29.5 ps to 23.2 ps. The Q-factor of PIT-III changed from 226 to 295.5, while the group delay increased from 78.1 ps to 105.7 ps. We can tell from the trend of the change in Q-factor and group delay that the Q-factor is proportional to group delay. Figure 9d suggests that the group delay of PIT-I shows the opposite trend to that group delay of PIT-IIII when we reduce the $E_{f}$. The results are consistent with our previous analysis that the Q-factor of PIT-I will increase caused by approaching to BIC point, while the Q-factor of PIT-III decreases because the coupling between bright mode-III and guided mode is becoming stronger. The comparison in Table 2 shows the excellent performance in slow light application. It is worth mentioning that the group delay in Ref. [43] was caused by a spoof localized surface plasmon interaction, and our design comes from the near-field coupling. The two mechanisms are slightly different, but both Ref. [43] and our device indicated that the group delay is related to the coupling strength. The Q-factor of the PIT spectrum is mainly related to the decay rate of the mode and the coupling strength. The slope of the transmission curves is inversely proportional to the square of the coupling coefficient. Therefore, the reduction of coupling strength will increase the Q-factor, which can lead to a larger group delay. In summary, the investigation of the slow light effect has revealed that our proposed structure can be applied for the optical delay in the terahertz range.

## 5. Conclusions

In this paper, we propose a micro-nano device based on the metamaterial-waveguided system, which can produce triple-PIT. Each of the PITs has its own character; in particular, the PIT-I can switch to BIC, and PIT-III is constructed by the coupling between low-Q wideband bright mode and high-Q narrowband guided mode. In addition, the primary resonance structure of dip 2 is graphene. Therefore, we can dynamically tune the triple-PIT by changing the $E_{f}$. A limitation of this study is that the tuning bandwidth is smaller than the devices with only patterned graphene, but we can still tune the triple-PIT’s Q-factor, which significantly influences the performance in sensing and slow light. On the one hand, the sensitivity *S* = 0.30 THz/RIU and the average FOM = 15.3 are achieved in this paper. On the other hand, the group delay can reach 29.5 ps for PIT-I when $E_{f}$ = 0.6 eV, while PIT-III’s group delay can reach 105.7 ps when $E_{f}$ = 1.0 eV. Therefore, our design has an excellent performance in slow light, and from the comparison with reported works, the refractive index sensor has good performance, and the tunable FOM is the prominent advantage of our design. The proposed devices also have potential applications in multi-frequency optical switching, narrow-band filtering, etc. Moreover, the design approach based on different mechanisms also provided a potential approach for designing the multi-PIT devices. 

## Figures and Tables

**Figure 1 nanomaterials-12-03273-f001:**
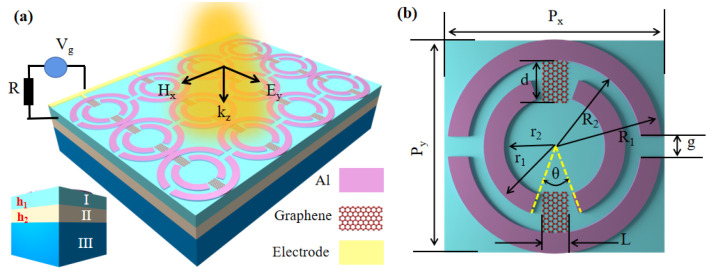
(**a**) Overall structure diagram; (**b**) schematic diagram of the top view structure of a unit.

**Figure 2 nanomaterials-12-03273-f002:**
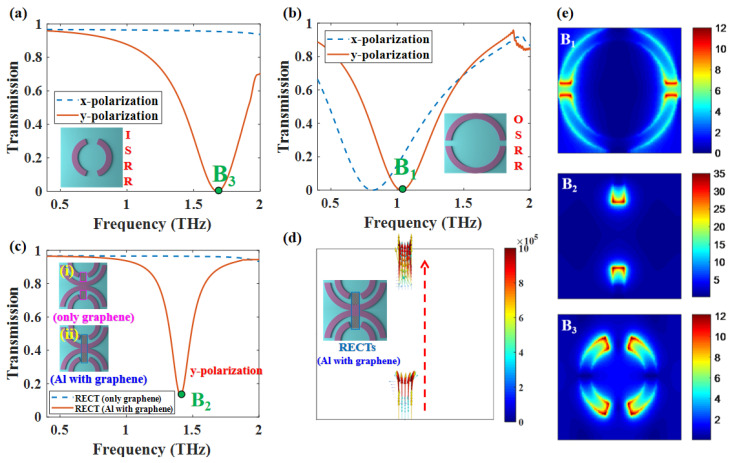
Γ = 0.3 meV, *N* = 1, *d* = 15 μm, $E_{f}$ = 1.1 eV. (**a**,**b**) The transmission curves of OSRR and ISRR with x/y-polarized normal incidence; (**c**) the transmission curves of RECTs (only graphene) and RECTs (Al with graphene); (**d**) The surface current of RECTs (Al with graphene) in 1.40 THz. (**e**) |*E*| field distributions of OSRR, ISRR, and RECTS.

**Figure 3 nanomaterials-12-03273-f003:**
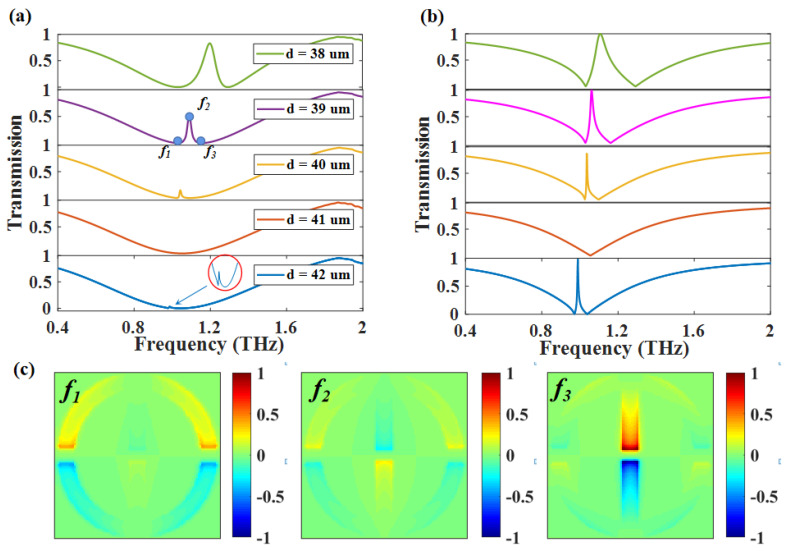
(**a**) The change in the transmission curve caused by changing geometric parameter *d*; (**b**) theoretical calculation of CMT without accounting for material losses; (**c**) $E_{z}$ electric field distribution corresponding to the three frequency points $f_{1}$, $f_{2}$, $f_{3}$ at *d* = 39 μm.

**Figure 4 nanomaterials-12-03273-f004:**
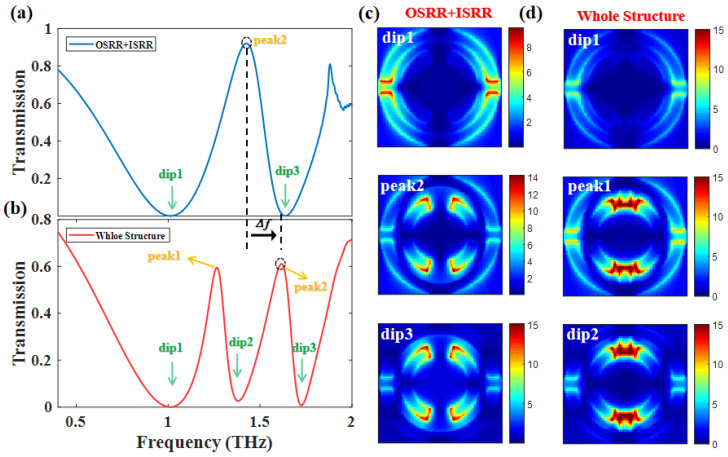
Γ = 0.3 meV, *N* = 1, *d* = 15 μm, $E_{f}$ = 1.1 eV. (**a**) Transmission spectrum of the PIT-II generated by coupling OSRR and ISRR (**b**) and the double-PIT formed with whole structure (OSRR+ISRR+RECTs); (**c**) electric field |*E*| distribution corresponding to the PIT-II; (**d**) electric field |*E*| distribution corresponding to the PIT-I.

**Figure 5 nanomaterials-12-03273-f005:**
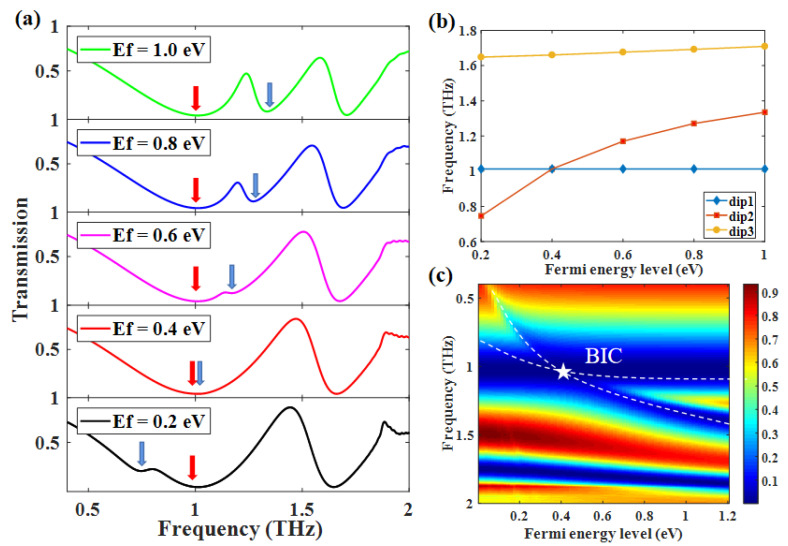
Γ = 0.3 meV, *N* = 1, *d* = 15 μm. (**a**) Dynamic tuning of double-PIT by changing the $E_{f}$ from 1.0 eV to 0.2 eV; (**b**) frequency shift of dip1, dip2, and dip3 as the $E_{f}$ changes; (**c**) Variation of transmission spectrum corresponding to different $E_{f}$.

**Figure 6 nanomaterials-12-03273-f006:**
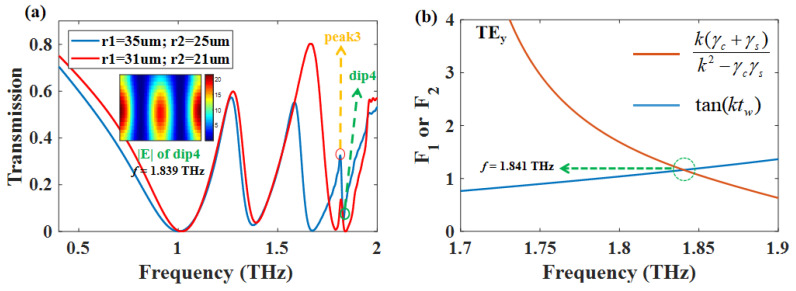
Γ = 0.3 meV, *N* = 1, *d* = 15 μm, $E_{f}$ = 1.1 eV. (**a**) Transmission curve change caused by changing the geometry of the ISRR; (**b**) the frequency of TE mode calculated by using the guided mode resonance condition; $F_{1}=\tan\left(kt_{w}\right)$ and $F_{2}=k\left(\gamma_{c}+\gamma_{s}\right)/\left(k^{2}-\gamma_{c}\gamma_{s}\right)$.

**Figure 7 nanomaterials-12-03273-f007:**
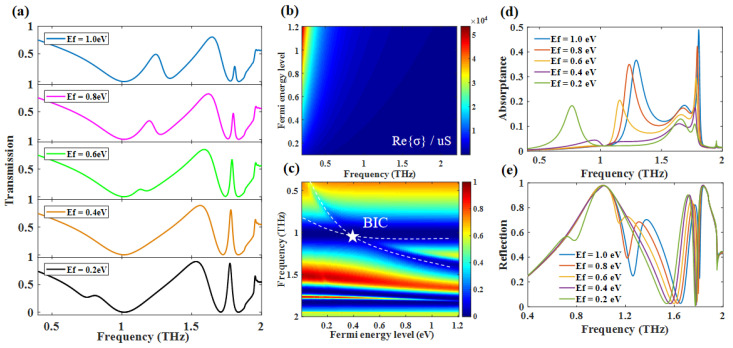
Γ = 0.3 meV, *N* = 1, *d* = 15 μm (**a**) Dynamic tuning of Triple-PIT by changing the $E_{f}$; (**b**) real part of graphene conductivity with different $E_{f}$; (**c**) transmission spectrum with different $E_{f}$; Changes in absorption (**d**) and reflection (**e**) curves caused by changing the $E_{f}$.

**Figure 8 nanomaterials-12-03273-f008:**
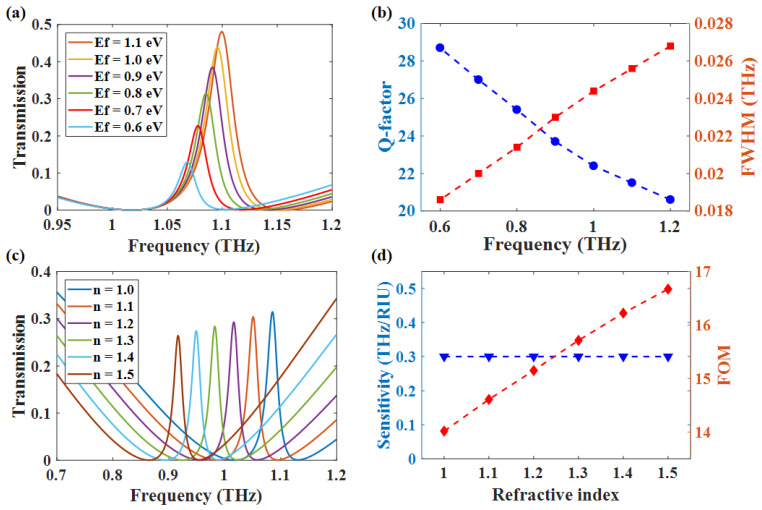
*N* = 7, *d* = 27 μm, μ = 4 $m^{2}/\mathrm{Vs}$ (**a**) The transmission curve of peak1 changes as $E_{f}$ decreases from 1.1 eV to 0.6 eV; (**b**) the change in Q value and FWHM due to the change in $E_{f}$; (**c**,**d**) the sensing sensitivity at $E_{f}$ = 0.8 eV and the FOM.

**Figure 9 nanomaterials-12-03273-f009:**
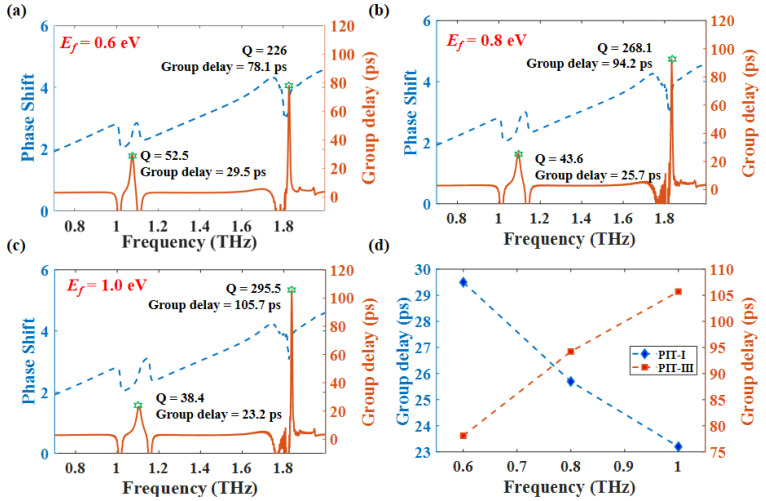
Γ=$1.37\times 10^{-4}$ eV, *N* = 7, *d* = 27 μm (**a**–**c**) Slow light effect of pek1 and pek3 at different $E_{f}$; (**d**) trends of changes in slow light effects caused by changes in $E_{f}$.

**Table 1 nanomaterials-12-03273-t001:** Comparison of the proposed sensor with other reported thz sensors.

	Our Proposed	Ref. [41]	Ref. [42]	Ref. [39]	Ref. [40]
S (THz/RIU)	0.30	0.16	0.13	1.48	1.96
FOM	16.6	1.04	62.8	24.6	19.86

**Table 2 nanomaterials-12-03273-t002:** Comparison of the proposed structure with others reported in slow light applications.

	Our Proposed	Ref. [16]	Ref. [43]	Ref. [7]
τ (ps)	29.5 or 105.7	10.22	42.4	5.74

## Data Availability

The calculated results during the current study are available from the corresponding author on reasonable request.

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
