# Peer review of "Multifunctional Plasmon-Induced Transparency Devices Based on Hybrid Metamaterial-Waveguide Systems"

_nanomaterials, 2022, doi:10.3390/nano12193273_

Round 1

Reviewer 1 Report

Chen et. al. has used numerical simulation and theoretical tool to show plasmon-induced transparency in a hybrid metamaterial-waveguide system. The work is interesting and well presented. This would be interesting for a broad scientific community that are interested in hybrid metamaterial-based devices. However, there are an ample of changes need to be considered before it can be considered for publication.

1.     The author mentioned in line 118 that the B2 resonance mode is due to Al and graphene provides bright mode and is supported with surface current. However, there was no enough evidence is given in order to support the claim apart from field plot. It would be interesting to see the surface current in z-direction in the heterojunction compared to Fig 2(iii).

2.     The author should include reflectance and absorbance curve together or separate as well. Transmission alone doesn’t explain everything. 

3.     The scale bar in transmission axis in figure 3 a and b doesn’t make any sense.

4.     Th author should mention the polarization orientation for both TM and TE and polarization dependent transmittance should be calculated for the OSSR, ISSR and hybrid structure.

5.     Figure 6b, y-axis labeling is missing.

6.     Is the simulation done with normal incidence of EM wave or oblique? If all are simulated at normal incident angle, the author should highlight what will happen for off-normal incident angle.

7.     I don’t understand why did the author choose fermi energy 0.8 eV as the best sample to do refractive index sensing. From figure 8a, it looks like 1.1 eV has maximum transmission. The author should highlight this

8.     While talking about RIU, how much the thickness of the active layer maters? Or it doesn’t matter here? Please explain.

9.     Compared to other literature the sensor performance is not great, please explain.

Reviewer 2 Report

The paper “Multifunctional plasmon-induced transparency devices based on hybrid metamaterial-waveguide systems” presents one more design for realization of triple plasmon-induced transparency device with graphene-assisted tuning. The paper is well written, seems to provide reasonable review and provides comparison of the proposed structure with others in application to refractive index sensing. Overall paper deserved to be published. There are, nevertheless, few things which in my view need to be corrected before publishing:

1.    1.   Figures are hard to read. Some lines almost not seen, number on axes are nearly impossible to read. I had to guess what results are. I suggest authors review all their figures and make sure axes are readable.

2.     2.  The suggested structure seems to be losing to the reference 41 by both refractive index sensitivity parameters. It would be nice, if authors would discuss a bit more this fact in the text and talk of advantages of the proposed scheme compared to one proposed in 41 if any. Similar discussion may need to appear in the conclusions.

3.       3. In conclusion: “Therefore, our design has excellent performance in refractive index sensing and slow light and it also has potential application in multi-frequency optical switching, narrow-band filtering, etc” – since there is paper with better performance, the refractive index performance is not excellent, it seems to be rather good. The only real advantage is on slow light performance. There may be some advantage in university of the device. I suggest to re-phrase.

4.     4.  The slow light performance seems to hold some record, but reference 43 gives comparable performance. It would be nice to give more detailed comparison with this device.

Round 2

Reviewer 1 Report

The author addressed majority of my concerns and it should be published in the present form.